

# Testing the potential of a ribosomal 16S marker for DNA metabarcoding of insects

Vasco Elbrecht[1,2], Pierre Taberlet[3,4], Tony Dejean[5], Alice Valentini[5], Philippe Usseglio-Polatera[6], Jean-Nicolas Beisel[7,8], Eric Coissac[3,4], Frederic Boyer[3,4] and Florian Leese[2,9]

[1] Department of Animal Ecology, Evolution and Biodiversity, Ruhr University Bochum, Bochum, Germany
[2] Aquatic Ecosystem Research, University of Duisburg-Essen, Essen, Germany
[3] Laboratoire d'Ecologie Alpine (LECA), CNRS, Grenoble, France
[4] Laboratoire d'Ecologie Alpine (LECA), Univ. Grenoble Alpes, Grenoble, France
[5] SPYGEN, Le Bourget du Lac, France
[6] Lab Interdisciplinaire des Environnements Continentaux (LIEC), Université de Lorraine, Metz, France
[7] Ecole Nationale du Génie de l'Eau et de l'Environnement de Strasbourg, Strasbourg, France
[8] UMR CNRS 7362—LIVE, Université de Strasbourg, Strasbourg, France
[9] Centre for Water and Environmental Research (ZWU) Essen, University of Duisburg-Essen, Essen, Germany

Corresponding author
Florian Leese, florian.leese@uni-due.de

## ABSTRACT

Cytochrome c oxidase I (COI) is a powerful marker for DNA barcoding of animals, with good taxonomic resolution and a large reference database. However, when used for DNA metabarcoding, estimation of taxa abundances and species detection are limited due to primer bias caused by highly variable primer binding sites across the COI gene. Therefore, we explored the ability of the 16S ribosomal DNA gene as an alternative metabarcoding marker for species level assessments. Ten bulk samples, each containing equal amounts of tissue from 52 freshwater invertebrate taxa, were sequenced with the Illumina NextSeq 500 system. The 16S primers amplified three more insect species than the Folmer COI primers and amplified more equally, probably due to decreased primer bias. Estimation of biomass might be less biased with 16S than with COI, although variation in read abundances of two orders of magnitudes is still observed. According to these results, the marker choice depends on the scientific question. If the goal is to obtain a taxonomic identification at the species level, then COI is more appropriate due to established reference databases and known taxonomic resolution of this marker, knowing that a greater proportion of insects will be missed using COI Folmer primers. If the goal is to obtain a more comprehensive survey the 16S marker, which requires building a local reference database, or optimised degenerated COI primers could be more appropriate.

## INTRODUCTION

DNA metabarcoding is a novel and powerful method to assess biodiversity in ecosystems (*Hajibabaei et al., 2011*; *Taberlet et al., 2012*; *Yu et al., 2012*; *Carew et al., 2013*; *Gibson et al., 2014*; *Leray & Knowlton, 2015*; *Dowle et al., 2015*). Well-designed universal PCR primers for the target group are the most critical component when assessing species diversity in ecosystems with DNA metabarcoding, because environmental samples typically contain hundreds of specimens of phylogenetically different taxa. Substantial primer bias in commonly used DNA barcoding markers, such as the Cytochrome c oxidase subunit I (COI) gene for animals, prevents the detection of all taxa in a sample and thus the estimation of taxa biomass is difficult (*Deagle et al., 2014*; *Piñol et al., 2014*; *Elbrecht & Leese, 2015*). However, accurate and comprehensive taxa lists are critical for assessment of biodiversity and ecosystem health. Given the great sequence variability of the COI marker, the use of alternative DNA metabarcoding markers has been suggested (*Clarke et al., 2014*; *Deagle et al., 2014*) and PCR-free metagenomics strategies are being tested for environmental assessment (*Gómez-Rodríguez et al., 2015*; *Tang et al., 2015*). One marker with potential for species level resolution and more conserved regions is the mitochondrial 16S rRNA gene (*Clarke et al., 2014*; *Deagle et al., 2014*). *Clarke et al. (2014)* has compared the performance of different COI and 16S primers on insect communities using an *in silico* approach which showed that the tested amplified ribosomal markers are generally more universal and detect more taxa than the COI markers. They also tested an insect mock sample containing DNA from 14 species with COI primers detecting the same amount or less taxa than with 16S. However, the performance of 16S metabarcoding primers with aquatic invertebrate communities has not been extensively tested.

In this study, we evaluated the performance of an insect primer pair targeting a short 16S region as compared to the standard COI Folmer marker (*Folmer et al., 1994*) for metabarcoding, using freshwater invertebrates mock communities. The ten freshwater mock communities were each comprised of 52 morphologically identified taxa and have been used in a previous study on COI primer bias (*Elbrecht & Leese, 2015*). Thus, they are ideal to comparatively evaluate the success rate of a short 16S fragment for DNA-based monitoring.

## MATERIAL AND METHODS

The same DNA aliquots as in *Elbrecht & Leese (2015)* were used to test the 16S marker to allow for a direct comparison. Laboratory conditions and bioinformatic analyses were kept as similar as possible to the study by *Elbrecht & Leese (2015)*.

### DNA metabarcoding

We used 16S markers ins_F/ins_R to amplify a ~157 bp of the mitochondrial 16S gene. This marker was developed as part of this project using the ecoPrimers program (*Riaz et al., 2011*) and represents a variant of the Ins16S_1short primer pair (*Clarke et al., 2014*). Fusion primers were used (Fig. S1), allowing to load PCR amplicons directly onto the Illumina NextSeq 500 sequencer. The same tag shifting and simultaneous sequencing

of forward and reverse primer and 10% PhiX spike in as described by *Lundberg et al. (2013)* and *Elbrecht & Leese (2015)* was used, to increase sequence diversity. Unique inline barcodes on forward and reverse reads were used for sample indexing.

The same one-step PCR and library preparation conditions as in *Elbrecht & Leese (2015)* were used with the following modifications: PCR extension time was reduced to 120 s and annealing temperature increased to 52.5 °C to better suit the fragment length and melting temperatures of the 16S Ins primers. Only one PCR replicate per sample was used for sequencing. Amplicons were purified with magnetic beads, but only a left-sided size selection was carried out to remove remaining primers and primer dimers (0.9x SPRIselect; Beckman Coulter, Bread, CA, USA). Concentrations were quantified with the Qubit BR Kit (Thermofisher Scientific, Carlsbad, CA, USA) and the library for sequencing was prepared by pooling 190 ng PCR product of all ten samples.

Paired-end Illumina sequencing was performed on a NextSeq 500 system using the mid output kit v2 kit with 300 cycles (150 bp PE sequencing) at the Alfred Wegener Institute Helmholtz Centre for Polar and Marine Research, Bremerhaven, Germany.

## Generation of 16S reference sequences

Due to the limited availability of 16S reference sequences on GenBank (NCBI), we constructed a reference library for the 52 morphotaxa used in this study, if tissue was still available. Standard DNA salt extraction, PCR, PCR clean-up, and Sanger sequencing were conducted as described in *Elbrecht et al. (2014)*, to amplify the 16S region with different primer sets and combinations. Primers were newly developed or checked for mismatches to Ephemeroptera, Plecoptera and Trichoptera using the PrimerMiner v0.2 R package (https://github.com/VascoElbrecht/PrimerMiner) and are available together with the generated reference sequences on BOLDsystems (TMIX Vasco). An annealing temperature of 52 °C was used for all primer combinations using HotMaster Taq (5Prime; Gaithersburg, Maryland, USA) for amplification.

## Bioinformatic analysis

Figure S2B includes a flow chart of the data processing steps. All used custom R scripts are available in Supplemental Information 1. First, reads were demultiplexed (R script splitreads_ins_v11.R) and paired end reads merged using USEARCH v8.0.1623 -fastq_mergepairs with -fastq_merge_maxee 1.0 (*Edgar & Flyvbjerg, 2015*). Primers were removed with cutadapt version 1.8.1 (*Martin, 2011*). Sequences from all ten replicates were pooled, dereplicated, and singletons were removed to find operational taxonomic units (OTUs) using the UPARSE pipeline (cluster_otus, 97% identity, *Edgar, 2013*). Chimeras were removed from the OTUs using uchime_denovo. The remaining OTUs were identified by querying against the nucleotide non-redundant database (NR) on NCBI using the Blast API (Entrez Programming Utilities) and our local 16S database using BLAST 2.2.31+ (*Camacho et al., 2009*). Taxonomy was assigned and checked manually, and in rare cases matches of ∼90% identity were accepted, if they matched the patterns which were previously reported for COI (*Elbrecht & Leese, 2015*).

The ten samples were dereplicated using derep_fulllength, but singletons were included in the data set. Sequences of each sample were matched against the OTUs with a minimum

match of 97% using usearch_global. The hit tables were imported and the sequence numbers were normalised to the total sequence abundance and tissue weight for the various taxa. Only OTUs with a read abundance above 0.003% in at least one replicate were considered in downstream analysis.

Due to the exponential nature of PCR, statistical tests on weight adjusted relative read abundances were carried out on decadic logarithm. Expected relative abundance was calculated by dividing 100% by the number of morphospecies detected with each marker.

## RESULTS

### Amplicon sequencing success and sequence processing

The NextSeq run generated 42.3 Gbp of raw sequencing data (NCBI SRA accession number SRR2217415). Cluster density was 177 K/mm$^2$ and read quality good with Q30 $\geq$ 85.3%. Read abundance was 17% higher when sequencing started with the P5_Ins_F primers ($t$-test, $p < 0.001$, Fig. S2A). This, however, did not introduce any significant differences between forward and reverse primer in the bioinformatic processing downstream ($t$-test, Fig. S2B).

Initial OTU clustering generated 855 OTUs of which 22.5% were detected as chimeras. Sequences from each sample were compared against the remaining 663 OTUs, but only 243 OTUs had at least one sample with >0.003% sequence abundance ($\sim$326 reads, SD = 29) and were thus included in further analysis. Taxonomy could be assigned for most OTUs based on available reference data and our own reference sequences. Reference data for the 16S marker could be generated for 42 of the 52 morphotaxa by Sanger sequencing. Together with 16S sequences from NCBI (Table S2) we were able to obtain reference data for all morphotaxa (Fig. 1). However, in several cases the NCBI data was obtained for morphotaxa identified at family or order level (e.g., Lymnaeidae, Nematoda, Acari Ceratopogonidae) did not yield species-level hits and thus were insufficient for reliable species identification. Table S1 gives an overview of assigned taxonomy for each OTU and Table S2 shows the distribution of detected taxa across the 10 replicates. Table S3 shows the sequence abundance for each morphotaxon from the 16S dataset as well as the COI dataset from *Elbrecht & Leese (2015)*, which was used for comparison of primer bias between both markers.

### Taxon recovery with 16S

The taxonomic assignment was straigthforward for the COI marker, due to the availability of reliable reference databases, which was not the case for the 16S marker. Forty-one out of 42 insect species were detected by the 16S. The Sanger sequence generated for the Tipulidae present in our mock samples showed 3 mismatches within the first bases at the 3′ end of both the forward and reverse 16S primers and was not detected in the data set.

### COI versus 16S

Most insect taxa were amplified with both markers, (38 out of 42), no insect taxon was only detected by COI, while 16S detected three more taxa (Ephemeridae, *Sericostoma personatum*, *Rhyacophyla*). The 16S primers worked very effectively for insect taxa,

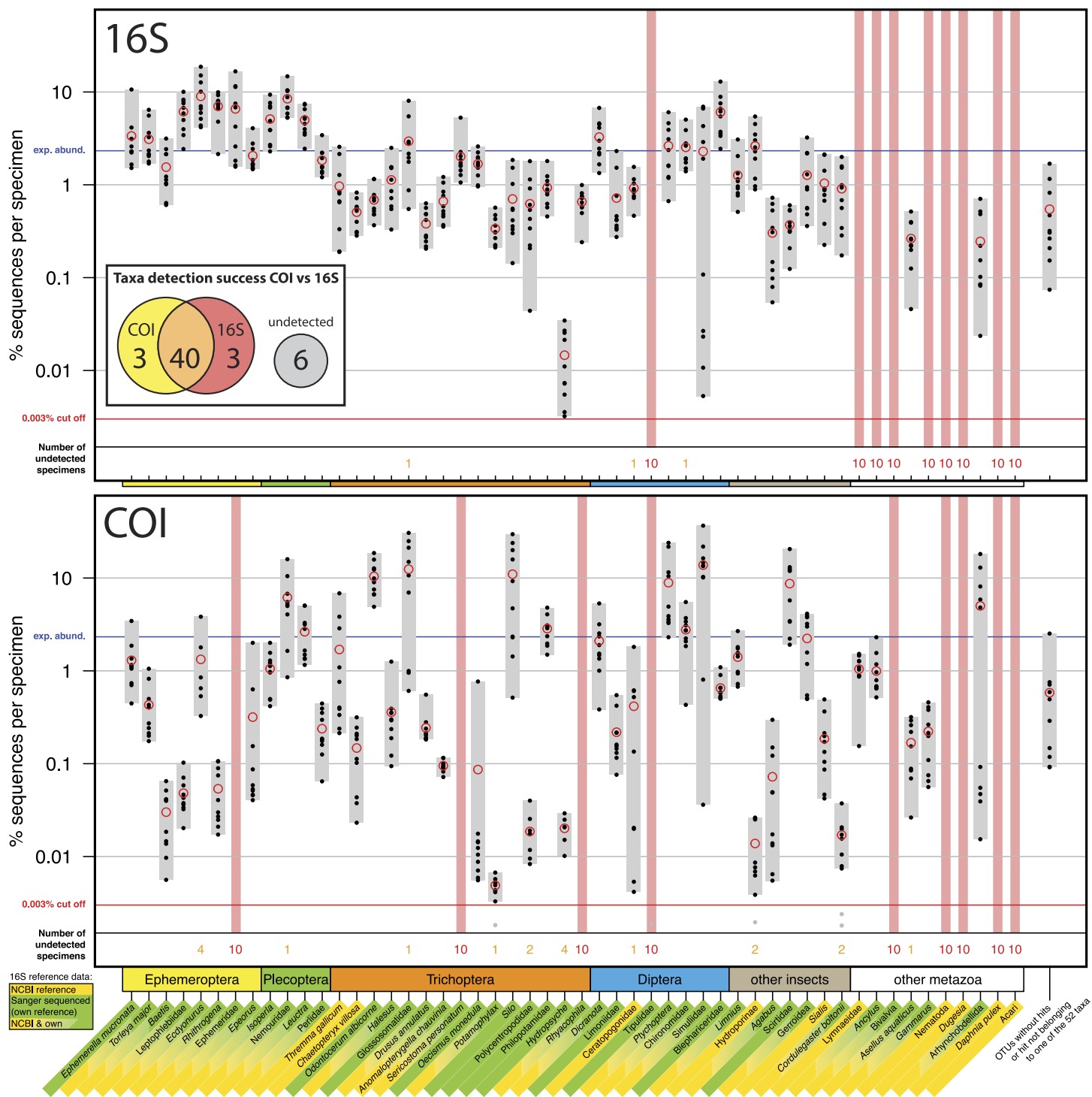

**Figure 1** Comparison of 16S Ins (A) and COI Folmer (B) primer performance, both tested with the same 10 bulk samples each containing 52 morphologically distinct macroinvertebrate taxa. Comparison of 16S Ins (A) and COI Folmer (B) primer performance, both tested with the same 10 bulk samples each containing 52 morphologically distinct macroinvertebrate taxa. (continued on next page...)

**Figure 1 (…continued)**
The 52 taxa are shown on the *x*-axis with the number of reads obtained with 16S and COI for each sample indicated by black dots on the logarithmic *y*-axis (mean relative abundance of detected morphotaxa is indicated by red circles). Sequence abundance was normalized across the ten replicates and the amount of tissue used in each DNA extraction. Only OTUs which had minimum abundance of 0.003% in at least one of the 10 samples were included in the analysis. Number of samples for which a morphotaxon was not detected is indicated by orange and red numbers in each plot. A thick vertical line in light red indicates if a morphotaxon was not detected. Detection rates between 16S and COI marker are summarized in a Venn diagram. The availability of 16S reference data from NCBI and own Sanger sequences is indicated by yellow and green background colour behind the taxon names on the *x*-axis.

**Table 1  Number of specimens recovered with the COI and 16S primers.**

| Taxonomic group | Recovered specimens | | | |
|---|---|---|---|---|
| | COI | | 16S | |
| Ephemeroptera | 7/8 | (88%) | 8/8 | (100%) |
| Plecoptera | 4/4 | (100%) | 4/4 | (100%) |
| Trichoptera | 13/15 | (86%) | 15/15 | (100%) |
| Diptera | 7/8 | (88%) | 7/8 | (88%) |
| Other insects | 7/7 | (100%) | 7/7 | (100%) |
| Other metazoa | 5/10 | (50%) | 2/10 | (20%) |
| Σ all insects | 38/42 | (91%) | 41/42 | (98%) |
| Σ all taxa | 43/52 | (83%) | 43/52 | (83%) |

specifically in the indicator taxa Ephemeroptera, Plecoptera and Trichoptera (100% detection success, Table 1). Of the ten other Metazoa, five were detected by COI, and only two by 16S. Variation in logarithmic insect read abundance was much lower for the ribosomal 16S amplicons (SD = 0.62%) than for the COI Folmer primers (SD = 1.0%) used on the same samples as in *Elbrecht & Leese (2015)* (Fig. 1). Logarithmic precision of relative read abundance (distance to expected abundance) was significantly better for 16S (SD = 0.37) than for COI (SD = 0.72, paired Wilcoxon signed-rank test, $p = 0.002$). Additionally, the COI primers showed more dropouts of a few specimens per taxa (orange numbers, Fig. 1), while the 16S primer with the exception of three cases always amplified all 10 specimens of a taxon. Table 1 compares the number of taxa recovered for the four most relevant orders for water quality assessment.

## DISCUSSION

We successfully ported our DNA metabarcoding protocol from the MiSeq system (*Elbrecht & Leese, 2015*) onto the NextSeq 500 that relies on sequencing by synthesis using only two instead of four channels for all four nucleotides (*Illumina, 2016*). As demonstrated in the previous study and also seen for the 16S Ins marker here, the use of fusion primers with a parallel sequencing strategy maximizes sequence diversity (see *Elbrecht & Leese, 2015*, Fig. S2), but can lead to a slight bias in read abundance. This however does not strongly affect read abundance of individual specimens between replicates (see Figs. S5 & S7 in *Elbrecht & Leese, 2015*). As in the previous test with COI Folmer primers, taxa not belonging to the 52 target taxa were detected with low abundances (<2% of data). This is likely a cause of

trace DNA, gut content or small overlooked tissue pieces in the extraction, in some cases possibly ambiguous hits due to low identity to matches in the NCBI/BOLD databases. However, these reads are not posing an issue as they were excluded in the analysis (OTUs without hits, Fig. 1). Further, a slight bias on sequence abundance might be introduced between and within samples by e.g., different amount of cuticula present when weighing tissue, tissue quality and variation in mitochondrial copy number. However, these effects are the same for both markers, so observed effects can be likely explained by primer bias. Here, we focus on comparing the results obtained from sequencing the mock community of 52 taxa using the two different markers and discuss their advantages and disadvantages.

## Power and limitations of 16S and COI markers in DNA metabarcoding

A key advantage of COI as a marker for DNA metabarcoding is that reference databases have been well established and are actively developed and extended (*Ratnasingham & Hebert, 2007*). DNA barcoding and the COI gene has been widely accepted by the scientific community as the barcoding marker of choice for animals (*Ratnasingham & Hebert, 2013*; *Porter et al., 2014*), despite some negative voices (*Taylor & Harris, 2012*). Additionally the taxonomic resolution of the COI marker has been extensively tested and its usefulness for identifying freshwater invertebrates on species level demonstrated (*Zhou et al., 2009*; *Pfrender et al., 2010*; *Zhou et al., 2010*; *Sweeney et al., 2011*). However, a documented concern of this marker is its large variability, which introduces primer bias due to mismatches at the primer binding sites (*Piñol et al., 2014*), which creates the risk of losing some target taxa (*Clarke et al., 2014*; *Deagle et al., 2014*). This large variation makes estimating biomass from PCR-based DNA metabarcoding results difficult (*Elbrecht & Leese, 2015*). The results of this study show that the 16S Ins primers show less amplification bias than the COI Folmer primers previously tested, which is coherent with previous results from *Clarke et al. (2014)*. Specifically for the Ephemeroptera, Plecoptera and Trichoptera, the 16S results were very consistent with variation in sequence abundance within these groups, with variation of only one order of magnitude for most taxa. A further advantage is that the reduced primer bias in 16S could allow for lower sequencing depths and thus a reduction in costs. The downside of using 16S as a marker at the present, however, is the limited availability of reference sequences and the yet not fully explored taxonomic resolution on species level. We had to establish our own 16S reference sequences for our mock communities *de novo* whenever tissue of the analysed morphotaxa was still available. This created extra work and cost that was omitted when using COI.

## Which marker to use?

COI is the standard marker for barcoding of animals and will typically yield the best resolved taxonomic lists. Therefore, if the goal of a project is to obtain a taxonomic identification at the species level, COI is more appropriate. However due to the codon degeneracy some taxa will likely not be amplified and thus missing in the dataset, making the COI marker not ideal when complete taxon lists are required. In direct comparison, three more insect taxa were not detected with the COI Folmer primers but found with the 16S Ins primers. Also *Clarke et al. (2014)* showed for various COI primers that they

performed either equally well or worse than the tested 16S primers. However, the use of improved COI primers with high degeneracy might lead to equally good detection and amplification consistency as with 16S, while allowing us to take advantage of existing COI reference databases. Degenerated COI primers like the "mlCOIint" primer sets by *Leray et al. (2013)* are being used in metabarcoding studies, but further evaluation concerning their primer bias is needed. Both the COI and 16S primer showed problems amplifying taxa listed in "other metazoa." This is expected for the Ins primers optimised for insect taxa and can be explained for the Folmer primers developed in the early 1990s with very little reference data available.

If the project goal is to obtain a more comprehensive survey and where it is possible to build a local reference database 16S can be a versatile and possibly even better alternative to COI, as this marker minimizes primer bias and provides more consistent PCR. Thus, 16S may possibly allow for rough biomass inferences, yet the variation of still two orders of magnitude as shown in this study show clear limitations as well. For species-level assignments, the potential of 16S remains largely unexplored for assessment of relevant invertebrate indicator taxa such as Ephemeroptera, Plecoptera and Trichoptera. Thus, prior to a routine application on 16S for species-level assessment we recommend reference sequencing of whole mitochondrial genomes using high throughput sequencing (*Tang et al., 2014*; *Coissac et al., 2016*), which not only allows for estimating taxonomic resolution of the two different mitochondrial markers, but also build the backbone for future metagenomic studies (*Tang et al., 2015*; *Coissac et al., 2016*). Once comprehensive mitochondrial reference databases are established, also direct PCR-free shotgun sequencing of bulk samples (metagenomics) becomes feasible. These approaches could further improve taxa detection rates and potentially allow to estimate taxa abundance (*Gómez-Rodríguez et al., 2015*; *Tang et al., 2015*). Using methods to enrich for mitochondrial reads we could further decrease sequencing costs for reference sequencing and mitogenomics approaches alike (*Zhou et al., 2013*; *Liu et al., 2015*; *Dowle et al., 2015*).

## CONCLUSIONS

In this study we show that the ribosomal 16S marker shows less primer bias than the COI barcoding marker with Folmer primers, when applied for DNA metabarcoding of freshwater insects. Thus, the developed 16S primers might allow to reduce sequencing depth in DNA based stream assessment, which could reduce sequencing costs. The main drawback when compared to COI is that little reference databases for stream invertebrates are available for the 16S marker and that taxonomic resolution remains largely unknown. This might change in the future when more reference data is generated, especially in the form of complete mitochondrial genomes generated with high throughput sequencing approaches. In cases where it is viable to generate local reference databases 16S could be a suitable alternative to COI. Additionally, degenerated COI primers should be evaluated as they are likely to perform better than the herein studied Folmer primers, which are not optimised for metabarcoding.

## ACKNOWLEDGEMENTS

We thank Uwe John and Nancy Kuehne (Alfred Wegener Institute Helmholtz Centre for Marine and Polar Research, Bermerhaven) for running the library on the MiSeq sequencer. We thank Laurence Clarke, Nathan Bott and one anonymous reviewer for helpful comments that substantially improved this study.

### Funding

Florian Leese and Vasco Elbrecht are supported by a grant from the Kurt Eberhard Bode foundation to Florian Leese. Pierre Taberlet, Eric Coissac, Tony Dejean, Alice Valentini, Philippe Usseglio-Polatera, Jean-Nicolas Beisel and Frederic Boyerwere were supported by a grant from Agence Nationale de la Recherche (aquaDNA; ANR-13-ECOT-0002-01). The funders had no role in study design, data collection and analysis, decision to publish, or preparation of the manuscript.

### Grant Disclosures

The following grant information was disclosed by the authors:
Kurt Eberhard Bode foundation.
Agence Nationale de la Recherche: ANR-13-ECOT-0002-01.

### Competing Interests

Tony Dejean and Alice Valentini are employees of SPYGEN.

### Author Contributions

- Vasco Elbrecht conceived and designed the experiments, performed the experiments, analyzed the data, wrote the paper, prepared figures and/or tables, reviewed drafts of the paper.
- Pierre Taberlet analyzed the data, wrote the paper, reviewed drafts of the paper, designed the ins 16S primers.
- Tony Dejean, Alice Valentini, Philippe Usseglio-Polatera, Jean-Nicolas Beisel and Frederic Boyer designed the ins 16S primers.
- Eric Coissac analyzed the data, reviewed drafts of the paper, designed the ins 16S primers.
- Florian Leese conceived and designed the experiments, performed the experiments, wrote the paper, reviewed drafts of the paper.

### DNA Deposition

The following information was supplied regarding the deposition of DNA sequences:
BOLDsystems (16S Sanger data): TMIX Vasco
SRA (Illumina data): SRR2217415

### Data Availability

Raw data has been supplied as Supplemental Information.

## Supplemental Information

Supplemental information for this article can be found online at http://dx.doi.org/10.7717/peerj.1966#supplemental-information.

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
