# Peer review of "Testing the potential of a ribosomal 16S marker for DNA metabarcoding of insects"

_PeerJ, doi:10.7717/peerj.1966_

## Round 0.1 · original submission · Minor Revisions

The reviewers were unanimous in their enthusiasm for the study. They raise a number of fairly small issues that can be dealt with in the course of a minor revision.

Reviewer 1 ·

Basic reporting

The text is clear, unambiguous and basic reporting accords with the requirements.

Experimental design

No Comments

Validity of the findings

L30 “Rough estimation of biomass might thus be less biased with 16S than with COI”. Based on table S3, it’s hard to agree with that statement. In general, PCR-based metabarcoding studies can’t make assessments about biomass.

Additional comments

This paper adds useful information about alternative metabarcoding primers for insects, also outlining the problems using 16S instead of COI.

How to explain the hits not belonging to the 52 mock taxa? For example, soil and litter associated collembolans found from streams and ponds?

Table S2. Does the nr 17 in cell D2 means that F0 primer (with R4?) resulted in 17 Ephemerella mucronata OTUs with 97% sequence similarity threshold?

The authors constructed a reference 16S library for the used morphotaxa. Was the barcoding gap tried to establish? Some taxa in Table S1 are conisdered as hits for the mock community based on 10% sequence dissimilarity? Isn't it too high for 16S?

L118 - what was the minimum number of reads per sample?
L 175 "Additionally the short length of the 16S marker used facilitates amplification, which is important when dealing with degraded DNA or eDNA.” - this applies also for COI mini-barcodes.
L 206 - too long sentence, hard to follow the idea.

·

Basic reporting

A little more information should be provided in the introduction to set the context for the study. Simlarly, a bit more info in the Material and Methods would make the manuscript more 'self-contained' (see comments below for both).

Experimental design

No comments

Validity of the findings

No comments

Additional comments

The study shows using a 16S metabarcoding marker improves the success rate for detection of insect taxa, with read abundances closer to the expected values compared to using the Folmer COI primers. Overall, it provides robust data and recommendations that should make a valuable contribution for the metabarcoding community.
My main criticism would be that the Folmer primers are somewhat of a ‘straw man’, in that several papers have already demonstrated the poor taxonomic coverage and PCR bias of these primers, including Clarke et al. (2014), Piñol et al. (2014), and the Elbrecht & Leese (2015) paper that this study builds upon. Although the current study provides a comparison with an alternative marker (16S), the performance of several COI and 16S primers for metabarcoding insects was compared in silico and in vitro by Clarke et al. (2014). The introduction should include a statement on how the current study differs from theirs.
Adding some information to the Materials and Methods would make this manuscript ‘stand-alone’ better from Elbrecht & Leese (2015). For example, ‘the mixtures were comprised of equal amounts of tissue (by dry weight) from 52 taxa’. I would also mention the number of PCR cycles (30?)
L81-82: Here and elsewhere the authors refer to the availability of 16S (and COI) reference sequences. Although it can be inferred from Figure 1, it would be useful to state the number of taxa for which reference data were available on GenBank/NCBI versus those where sequences needed to be generated de novo, especially for 16S, if not for COI as well.
L130-131: I guess you cannot comment on the presence of mismatches for the metazoans, especially those that you couldn’t generate Sanger sequences.
L183-185: It would be useful to refer to the results of the study here. For example, providing the two ranges of the number of the 42 insect taxa not detected in each replicate for COI versus 16S would indicate the percentage of taxa that might be missed in metabarcoding studies using the Folmer primers alone.
L186-187: Many degenerate COI primers are already being used for metabarcoding, e.g. Leray et al. (2013) Frontiers in Zoology 10:34. It should be made clear that the efficacy of these primers really needs to be tested, as they may provide the combination of reference data and reduced PCR bias needed for the successful application of DNA metabarcoding.
L189-192: It seems to me that, depending on the available sequence data, the accuracy of species-level assignments for Ephemeroptera, Plecoptera and Trichoptera could be easily established by in silico PCR, which would be much less expensive and time-consuming than mitochondrial genome sequencing.
Minor comments
L42: Suggest ‘well-designed universal PCR primers for the target group are…’
L97: What is ‘Blast API’?
L98-100: The percentage identities could be supplied in a supplementary table for transparency.
L154: Does ‘dual sequencing strategy’ refer to paired-end sequencing?
L202: Make it clear that you found less primer bias for one 16S marker compared to one COI marker, as this may not be the case for degenerate COI primers.
Figure 1. The Venn diagram could be redrawn to provide a better visual indication that the majority of taxa were detected with both markers.

·

Basic reporting

Elbrecht et al discuss the potential for the use of ribosomal 16S marker for the DNA metabarcoding of insects. The manuscript is generally well conceived and written. I only have minor comments about the manuscript.

Line 63: The used 16S markers seems a strange use of words. Suggest change to We used. Also were those primers developed as part of this study or from a previous study? It is not entirely clear.

Line 72: what brand of magnetic beads? AmPure or another?

Line 82: Change to the 52 morphotaxa used in this study

Line 130: change to present. Also is it possible that the mismatches are due to sequencing error, or is there sequence heterogeneity known for this gene?

Line 152: This is the first time the authors discuss MiSeq vs NextSeq, I suggest they either incorporate this into the introduction or re-word here.

Line 180: remove extra period

Line 182: remove at the present

Line 196: These instead of this

Experimental design

Well designed and entirely appropriate for the nature of the study.

Validity of the findings

Findings are valid.

Conclusions are appropriate. It would also be useful if the authors discussed in some more detail the poor recovery of other metazoans (i.e. Table 1). The comments about sequencing mitogenomes are salient points and worthy of further investigation.

---

## Round 0.2 · accepted · Accept

I believe that all the authors deal with all of the minor changes satisfactorily.